# Essential Trace Elements Status in Portuguese Pregnant Women and Their Association with Maternal and Neonatal Outcomes: A Prospective Study from the IoMum Cohort

**DOI:** 10.3390/biology12101351

**Published:** 2023-10-21

**Authors:** Isabella Bracchi, Juliana Guimarães, Catarina Rodrigues, Rui Azevedo, Cláudia Matta Coelho, Cátia Pinheiro, Juliana Morais, Inês Barreiros-Mota, Virgínia Cruz Fernandes, Cristina Delerue-Matos, Edgar Pinto, André Moreira-Rosário, Luís Filipe Ribeiro de Azevedo, Cláudia Camila Dias, Jorge Lima, Inês Sapinho, Carla Ramalho, Conceição Calhau, João Costa Leite, Agostinho Almeida, Diogo Pestana, Elisa Keating

**Affiliations:** 1Unit of Biochemistry, Department Biomedicine, Faculty of Medicine, University of Porto, 4200-319 Porto, Portugal; 2CINTESIS, Center for Health Technology and Services Research, 4200-319 Porto, Portugal; 3CHRC, NOVA Medical School, Faculdade de Ciências Médicas, NMS, FCM, Universidade NOVA de Lisboa, 1150-082 Lisbon, Portugal; 4LAQV/REQUIMTE, Department of Chemical Sciences, Faculty of Pharmacy, University of Porto, 4050-313 Porto, Portugal; 5CINTESIS@RISE, Nutrition & Metabolism, NOVA Medical School|FCM, NOVA University Lisbon, 1169-056 Lisbon, Portugal; 6UniC@RISE, Unidade de Investigação e Desenvolvimento Cardiovascular, Faculdade de Medicina, Universidade do Porto, 4200-319 Porto, Portugal; 7REQUIMTE/LAQV, ISEP, Polytechnic of Porto, rua Dr. António Bernardino de Almeida, 4249-015 Porto, Portugal; 8Departamento de Saúde Ambiental, Escola Superior de Saúde, Instituto Politécnico do Porto, Rua Dr. António Bernardino de Almeida 400, 4200-072 Porto, Portugal; 9CINTESIS@RISE, Department of Community Medicine, Information and Health Decision Sciences (MEDCIDS), Faculty of Medicine, University of Porto, 4200-319 Porto, Portugal; 10Immunology, NOVA Medical School, Faculdade de Ciências Médicas, NMS, FCM, Universidade NOVA de Lisboa, 1169-056 Lisboa, Portugal; 11Department of Obstetrics and Gynecology, Hospital da Luz Lisboa, 1500-650 Lisboa, Portugal; 12Endocrinology Service, CUF Descobertas Hospital, 1998-018 Lisbon, Portugal; 13Department of Obstetrics, São João Hospital Center, 4200-319 Porto, Portugal; 14Department of Ginecology-Obstetrics and Pediatrics, Faculty of Medicine, University of Porto, 4200-319 Porto, Portugal; 15Institute for Research and Innovation in Health, i3S, University of Porto, 4200-135 Porto, Portugal; 16CINTESIS@RISE, Faculty of Medicine, University of Porto, 4200-319 Porto, Portugal

**Keywords:** essential trace elements, pregnancy, pregnancy complications, pre-eclampsia, neonatal outcomes, neonatal anthropometry

## Abstract

**Simple Summary:**

Cobalt (Co), copper (Cu), manganese (Mn), molybdenum (Mo), and zinc (Zn) are essential trace elements (ETEs) important in cellular chemical reactions and antioxidant defense. Ingestion of ETEs during pregnancy is crucial but their role in specific pregnancy outcomes is largely unknown. This study aimed to quantify urinary levels of these ETEs in pregnancy and to evaluate their role in pregnancy health. First trimester pregnant women of Porto and Lisbon regions provided a urine sample, and sociodemographic and lifestyle data. Clinical data were obtained from clinical records. Urinary ETEs were quantified by inductively coupled plasma mass spectrometry (ICP-MS). Our results show that having urinary Zn levels above the 50th percentile (P50) increases the risk of pre-eclampsia (PE). On the other hand, urinary Zn levels above the P50 decreased the risk of being born with head circumference small for gestational age but it increased the risk having length small for gestational age at birth. This study may provide valuable information for public health policies related to prenatal nutrition, while informing future efforts to de-fine urinary reference intervals for ETEs in pregnant women.

**Abstract:**

Cobalt (Co), copper (Cu), manganese (Mn), molybdenum (Mo), and zinc (Zn) are essential trace elements (ETEs) and important cofactors for intermediary metabolism or redox balance. These ETEs are crucial during pregnancy, their role on specific pregnancy outcomes is largely unknown. This prospective study (#NCT04010708) aimed to assess urinary levels of these ETEs in pregnancy and to evaluate their association with pregnancy outcomes. First trimester pregnant women of Porto and Lisbon provided a random spot urine sample, and sociodemographic and lifestyle data. Clinical data were obtained from clinical records. Urinary ETEs were quantified by inductively coupled plasma mass spectrometry (ICP-MS). A total of 635 mother:child pairs were included. Having urinary Zn levels above the 50th percentile (P50) was an independent risk factor for pre-eclampsia (PE) (aOR [95% CI]: 5.350 [1.044–27.423], *p* = 0.044). Urinary Zn levels above the P50 decreased the risk of small for gestational age (SGA) birth head circumference (aOR [95% CI]: 0.315 [0.113–0.883], *p* = 0.028), but it increased the risk SGA length (aOR [95% CI]: 2.531 [1.057–6.062], *p* = 0.037). This study may provide valuable information for public health policies related to prenatal nutrition, while informing future efforts to de-fine urinary reference intervals for ETEs in pregnant women.

## 1. Introduction

Pregnancy corresponds to a metabolically challenging period of continuous anatomical and physiologic adjustments [1]. The relevance of micronutrients such as folic acid, iron (Fe) and iodine to support all fundamental biological processes and to ensure maternal and offspring health is well described [2]. Meanwhile, interest in other essential trace elements (ETEs) in pregnancy has been notoriously emerging [1].

According to the classification proposed by Frieden [3] and later by the Expert Consultation of World Health Organization (WHO)/Food and Agricultural Organization (FAO)/International Atomic Energy Agency (IAEA) [4], ETEs, or microminerals, are minerals required in amounts less than 100 mg/day or which account for less than 0.01% of total body mass. Examples of ETEs are cobalt (Co), copper (Cu), manganese (Mn), molybdenum (Mo), and zinc (Zn) [3,4]. These elements are available in food sources in association with proteins but they also occur in inorganic forms, for example in multivitamin multimineral nutritional supplement formulations. These ETEs play important roles as cofactors in various enzymatic and antioxidant processes and they are crucial for sustaining diverse metabolic processes [5] (Table 1).

Metabolic and physiological adjustments during pregnancy globally require an increased intake of several ETEs. Inadequate systemic levels of ETEs during pregnancy, whether too low or too high, are associated with poor pregnancy and neonatal outcomes [1,25,26].

For example, low maternal Zn or Co levels have been reported to be associated with an increased risk of hypertensive disorders of pregnancy [27,28] and maternal and umbilical cord Zn concentrations have been shown to be positively associated with birth weight [29]. Additionally, increased maternal blood levels of Mo and Mn during pregnancy have been shown to be protective against abnormal glucose levels [30] and intrauterine growth restriction [31], respectively. On the other hand, despite the nutritional relevance of Cu, higher maternal plasma Cu levels in early pregnancy (12 weeks of gestation) were associated with an increased risk of maternal glucose dysregulation [30]. Significantly higher umbilical cord blood Cu levels were found in neonates with small-for-gestational age birth weight [29].

Despite the large number of studies that highlight the importance of ETE adequacy during pregnancy, the literature is scarce in the definition of reference values for ETEs, particularly in urine [32]. Urine is an easy-to-obtain and non-invasive biological sample that offers the possibility of carrying out large-scale monitoring studies. There is also a lack of epidemiological data on social determinants and lifestyle factors that may influence urinary ETEs levels in the pregnant population, especially in Europe.

Regarding the Portuguese population, there are no studies that address this topic, and the determination of Co, Cu, Mo, Mn, and Zn urinary levels and the possible association with maternal and neonatal outcomes have not been studied yet.

Understanding the role of ETEs status in pregnant women is important for optimizing both maternal and child health throughout the lifespan. Furthermore, the characterization of ETEs status during pregnancy can provide valuable information for public health policies related to prenatal nutrition.

Thus, the objectives of the present study were: (a) to evaluate the urinary levels of Co, Cu, Mn, Mo, and Zn in Portuguese pregnant women during the 1st trimester of pregnancy, and (b) to evaluate their association with maternal data and with pregnancy and neonatal outcomes.

## 2. Materials and Methods

### 2.1. Ethical Approval

This study relies on the IoMum project—Iodine Status Monitoring in Portuguese Pregnant women: Impact of Supplementation (Clinical Trials: NCT04010708) which was carried out according to the protocol approved by the Ethics Committee of Centro Hospitalar e Universitário de São João (CHUSJoão)/Faculty of Medicine of the University of Porto and Hospital CUF Descobertas, in Lisbon. Written informed consent was obtained from all subjects.

### 2.2. Study Design and Population

This is a prospective observational study, carried out from the IoMum pregnancy cohort.

Pregnant women who underwent their first trimester routine fetal ultrasound at CHUSJoão, Porto, Portugal, between April 2018 and April 2019, and at Hospital CUF Descobertas, Lisbon, Portugal, between October 2020 and December 2021 were invited to participate. At this stage, participants completed a structured questionnaire that collected information on lifestyle, anthropometric, sociodemographic, and dietary data. Dietary information was collected through semi-quantitative food intake questions addressing the intake of milk, dairy, fish and seafood, and offal. Some dietary information exists only for the Lisbon sub-cohort due to differences in questionnaire questions between sub-cohorts.

Additionally, a random spot urine sample was collected. More information about pregnancy and neonatal outcomes, as well as neonatal data, was obtained from clinical records.

All pregnant women enrolled in the IoMum study who signed the informed consent and confirmed fetal vitality at recruitment were included. Women who dropped out from pregnancy surveillance at one of the recruitment hospitals and for whom we did not obtain delivery and neonatal anthropometric data, and women with twin pregnancy, gestational age outside the 1st trimester range at recruitment (<10 or ≥14 weeks) and who did not deliver urine samples were excluded. Afterwards, this study covered a total of 635 participants, as illustrated in Figure 1.

### 2.3. Biochemical Analyses

At the end of each recruitment day, urine samples were aliquoted and stored frozen at −80 °C. The determination of the urinary concentration of ETEs was performed by inductively coupled plasma mass spectrometry (ICP-MS), which is the most appropriate technique for this purpose [33]. The ICP-MS instrument was an iCAP™ Q (Thermo Fisher Scientific, Bremen, Germany), equipped with a Meinhard^®^ (Golden, CO, USA) TQ+ quartz concentric nebulizer, a Peltier cooled, high purity quartz, baffled cyclonic spray chamber and a demountable quartz torch with a 2.5 mm i.d. quartz injector. The interface consisted of two (sampler and skimmer) Ni cones. High-purity argon (99.9997%) supplied by Gasin (Leça da Palmeira, Portugal) was used as the nebulizer and plasma gas. Before each analytical run, the instrument was tuned for maximum sensitivity and signal stability and for minimal formation of oxides and double-charged ions. The main parameters of ICP-MS operation were as follows: nebulizer gas flow, 1.13 L/min; auxiliary gas flow, 0.79 L/min; plasma gas flow, 13.9 L/min; power of radio frequency generator, 1550 W; dwell time, 10–30 ms.

The analytical procedure used was based on laboratory protocol described by the Centers for Disease Control and Prevention (CDC) (Laboratory Procedure Manual, Method 3018.3) [34]. Urine samples were diluted 1:10 with a diluent solution containing 2% *v*/*v* HNO_3_, 1.5% *v*/*v* ethanol, 500 µg/L Au (Gold Standard for ICP, 1000 mg/L, TraceCERT^®^, Sigma-Aldrich, St. Louis, MO, USA) and internal standards (IS) at 10 µg/L (Periodic table mix 3 for ICP, 10 mg/L, TraceCERT^®^, Sigma-Aldrich). A rinse solution of 5% *v*/*v* HNO_3_, 1.5% *v*/*v* ethanol, 0.002% *v*/*v* Triton X-100 and 500 µg/L Au was pumped into the sample introduction system between each sample to prevent carry-over of the analytes from one sample measurement to the next.

Matrix-matched standard solutions were used for the external calibration of each analytical run. A urine pool was created from stored urine samples analyzed in previous research projects and added (1:10 dilution) during the preparation of every standard. For the determination of Mn, Co, and Cu, an 8-point calibration curve (1, 5, 10, 25, 50, 100, 250, and 500 µg/L) was generated with standard solutions prepared by adequate dilution of the multi-element solution Periodic table mix 1 (10 mg/L, TraceCERT^®^, Sigma-Aldrich). For Zn, a 13-point calibration curve (0, 1, 5, 10, 25, 50, 100, 250, 500, 1000, 1500, 2000, and 5000 µg/L) was generated with standard solutions prepared by adequate dilution of the multi-element solution Periodic table mix 1 and a single-element Zn solution (Zinc standard for AAS, 1000 mg/L, SCP Science, Quebec, QC, Canada). For Mo, a 10-point calibration curve (0, 0.05, 0.25, 0.5, 1.25, 2.5, 5, 12.5, 25 and 100 µg/L) was generated with standard solutions prepared by adequate dilution of the multi-element solution Periodic table mix 2 (10 mg/L, TraceCERT^®^, Sigma-Aldrich). These calibration solutions were diluted 1:10 with the diluent solution, as were the samples. The elemental isotopes 55Mn, 59Co, 65Cu, 66Zn and 98Mo were measured for analytical determinations and the elemental isotopes 45Sc, 89Y, 141Pr, and 175Lu were monitored as IS.

A matrix-matched blank was prepared daily with the same urine pool used for the preparation of the standards. The matrix-matched blank was analyzed 10 times for the daily automatic estimation of the method limit of determination (LOD) by the instrument’s software (Qtegra, version 2.14.5122.158, Thermo Fisher Scientific). For each ETE LOD was determined and results below LOD were imputed as LOD/√2, as recommended by the German Human Biomonitoring Commission (HBM Commission) [35]. These values were then considered for all statistical analyses.

For analytical quality assurance, repeated analysis (at the beginning, middle, and end of each analytical run) of Seronorm™ Trace Elements Urine L-1 and L-2 (obtained from SERO AS, Billingstad, Norway) was performed.

#### Creatinine Determination

Urinary creatinine determination was performed by certified laboratories using the Jaffe’s reaction method. For creatinine adjustment, the ETEs-to-creatinine ratios (μg/g) were calculated by dividing ETEs’ urinary concentrations in μg/L by the concentration of creatinine in g/L.

### 2.4. Variables

#### 2.4.1. Categorization of Anthropometric Percentiles at Birth

Anthropometric percentiles at birth were calculated based on the revised Fenton-2013 growth chart [36] and international standards for weight, length, and head circumference [37], taking into consideration the gestational age at delivery. Subsequently, three different percentile categories were created for each anthropometric parameter: small-for-gestational-age (SGA; <10th percentile), adequate for gestational age (AGA; ≥10 to 90th ≤ percentile) and large for gestational age (LGA; >90th percentile) [38].

#### 2.4.2. Classification of Gestational Weight Variation

The variation in gestational weight was calculated as follows:Total weight variation = maternal weight at term – maternal weight 6 months before pregnancy (1)
1st trimester weight variation = maternal weight at recruitment day – maternal weight 6 months before pregnancy (2)
where maternal weight at term was obtained from the clinical records and both maternal weight at recruitment day and maternal weight 6 months before pregnancy were reported by the participant in the applied structured questionnaire.

Afterwards, the adequacy of weight variation during the 1st trimester and throughout the entire pregnancy was categorized according to international guidelines for weight gain during pregnancy [39]. Thus, the classification of weight variation was established as: below adequate (weight loss or weight gain less than the recommended), adequate (weight gain as recommended), and above adequate (weight gain greater than recommended).

#### 2.4.3. Categorization of Supplementation Use

The use of food supplements was evaluated according to what was reported by the participants in the questionnaires. When the food supplement contained at least two vitamins or minerals, it was considered a multivitamin/multimineral formulation (MV/MMF). When the food supplement contained only one vitamin or mineral (e.g., iodine or folic acid), it was considered a single ingredient formulation (SIF). Composition of different MV/MMF and SIF reported to be used by participants is presented in Appendix A.

### 2.5. Statistical Analysis

The normality of continuous variables was assessed using the Kolmogorov–Smirnov test and by visual inspection of the histograms. For variables with non-normal distribution, the results were presented as median and interquartile range (IQR—25th–75th percentile). For variables with normal distribution, the results were presented as means and standard deviation (SD). Categorical variables were presented as absolute and relative frequencies.

When testing hypotheses about continuous variables, parametric test (Student’s *t*-test) or nonparametric tests (Mann–Whitney and Kruskal–Wallis) were used according to the assumed normality and the number of groups evaluated. Additionally, to analyze independence between two categorical variables, the Pearson chi-square test was used.

Multivariate logistic regression models were performed to analyze the risk for pregnancy complications and SGA and LGA anthropometry, according to ETEs urinary levels. These models were applied to independent and dependent variables that revealed to be significantly associated.

For multivariate logistic regressions, the 50th percentile of urinary Zn concentration (256.9 µg/L) or the 35th and the 60th percentiles of urinary Mn concentration (1.02 and 2.11 µg/L) were set as cut-off points based on the statistical significance of the associations found between urinary Zn or Mn concentrations and pregnancy or neonatal outcomes (percentile values for urinary Zn and Mn concentrations are provided in Appendix A).

The following covariates were considered given their biological relevance for the outcomes being measured: sub-cohort, gestational weight gain, educational level, smoking habits, maternal body mass index (BMI), maternal age, parity, newborn’s sex, and pregnancy complications. The adjusted odds ratio (aOR) and the corresponding 95% confidence intervals were presented. Statistical significance was assumed when a 2-tailed *p* value < 0.05 was obtained.

Data analyses were performed using the statistical software SPSS^®^ v.27.0 (Statistical Package for the Social Sciences) and box plot graph was created using GraphPad^®^ Prism software version 9.5.1.

## 3. Results

### 3.1. Study Sample Characterization

The sociodemographic data of the 635 pregnant women included in this study are shown in Table 2. Fifty-three percent of the study sample was from the Porto sub-cohort. The overall mean (SD) age was 33 (5) years and the median (P25; P75) gestational age at recruitment was 12 (12; 13) weeks. Almost half of the women (46%) had a university degree, 30% were smokers or ex-smokers, and 46% used MV/MMF food supplements. Only 4% of the study sample did not take any type of food supplement.

Regarding anthropometric characteristics, 63% percent of the pregnant women started pregnancy with normal weight; however, only 31% had adequate weight gain in the 1st trimester, while 48% gained more weight than the recommended, in the 1^st^ trimester. Total gestational weight gain followed the same trend.

Twenty percent of pregnant women were diagnosed with at least one pregnancy complication (including preeclampsia, gestational diabetes, fetal growth restriction, infection, preterm birth, and spontaneous fetal loss) and the most prevalent complication was gestational diabetes mellitus, affecting 9% (*n* = 56) of the whole cohort. In addition, 51% of pregnant women were nulliparous.

Regarding the characteristics of newborns, 51% were female, and the median (P25; P75) head circumference at birth was 34.0 (33.5; 35.0) cm. Mean (SD) weight and length at birth were 3207 (433) g and 49 (2) cm, respectively. The vast majority were categorized as AGA for birth weight (92%), birth head circumference (86%) and birth length (92%).

When comparing sociodemographic data between sub-cohorts, statistically significant differences were found for maternal age (*p* < 0.001), education (*p* < 0.001), smoking habits (*p* = 0.001), use of food supplements (*p* < 0.001), pregnancy complications (*p* = 0.041), and newborns weight (*p* = 0.002), head circumference (*p* < 0.001), and length (*p* = 0.029) at birth.

### 3.2. ETEs Levels and Association with Sociodemographic Characteristics

Maternal urinary ETEs concentrations are summarized in Table 3 (*n* = 635). All samples had detectable Cu and Mo, while 8 and 18% of the samples had Mn and Co levels below the detection limits, respectively. Only 1 participant had Zn below the corresponding LOD.

When exploring the association of urinary ETEs concentrations with sociodemographic characteristics (Table 4), differences were found for almost all variables.

Regarding sub-cohorts, 2- and 4-times higher concentrations of Cu and Mn (*p* < 0.001), respectively, were found in Porto compared with Lisbon sub-cohort.

Pregnant women with lower educational levels had higher urinary levels of Cu (*p* < 0.001) and Mn (*p* <0.001). On the contrary, urinary Mo levels increased with increasing educational level (*p* = 0.007). Smoking mothers had significantly higher levels of Cu and Mn when compared with non-smoking mothers (*p* = 0.047 and *p* < 0.001, respectively). The same was observed for Co, but this association was significant only when Co levels were adjusted for urinary creatinine (Appendix A).

Cu levels increased with increasing pre-pregnancy BMI, with the highest median values among obese pregnant women (*p* = 0.027). In addition, pregnant women who gained more weight than recommended for the 1st trimester had higher urinary Co levels compared with pregnant women with adequate or less than adequate gestational weight gain (*p* = 0.020). Furthermore, pregnant women who had adequate weight variation in the 1st trimester had significant lower Cu levels compared with pregnant women with variation below or above the recommended (*p* = 0.008).

No statistically significant difference was observed in maternal urinary concentrations of ETEs (with or without adjustment for creatinine) as a function of newborn sex.

### 3.3. ETEs Status in Association with Food Intake and Food Supplementation

Data on MV/MMF use and its association with maternal ETEs urinary levels are provided in Figure 2. None of the pregnant women included in the study reported the use of Mo-containing supplements, so this element was excluded from this analysis.

When analyzing the urinary concentrations of each individual ETE and the use of MV/MMF, urinary Cu and Mn levels were significantly lower in pregnant women using MV/MMF (*p* < 0.001 for both ETEs). On the other hand, urinary Co levels were similar in MV/MMF users compared with non-users (0.35 μg/L vs. 0.34 μg/L; *p* = 0.062) and urinary Zn levels tended to be lower in MV/MMF users, but the difference was not statistically significant (*p* = 0.123).

Table 5 shows the association between food intake and urinary levels of ETEs. Mn levels increased for higher frequencies of milk consumption (*p* = 0.002) and this was also the case when Mn levels were adjusted for creatinine excretion (Appendix A). Increasing urinary levels of Cu or Mn adjusted for creatinine were also observed for higher consumptions of yoghurt (Appendix A). On the contrary, decreasing urinary levels of Cu and Mn were observed for increasing consumption of fish and eggs.

Similarly, urinary levels of Mo and Zn were lower for higher frequencies of offal consumption (*p* = 0.046 and 0.047, respectively).

No further statistically significant associations were found between consumption of yoghurt, cheese, seafood, legumes, or nuts and the studied ETEs.

### 3.4. Association between ETEs, Pregnancy Outcomes and Neonatal Anthropometric Data

Differences in urinary ETEs levels between uncomplicated and complicated pregnancies were analyzed.

Levels of Zn above the 50th percentile (256.9 µg/L) were observed among women who had at least one pregnancy complication compared with women who had no pregnancy complications (*p* = 0.048) (Figure 3). This was also the case when urinary Zn levels were adjusted for creatinine excretion, but the difference was no longer statistically significant (Appendix A). When analyzing each individual complication, no statistical significance was found for Zn, but there was a trend towards higher levels of Zn among pregnant women diagnosed with preeclampsia (PE) compared with healthy pregnant women (398.0 µg/L vs. 253.2 µg/L; *p* = 0.067).

Additionally, lower or higher levels of urinary Mn were associated with specific complications (Figure 4). Urinary Mn levels below the 35th percentile (1.02 µg/L) were associated with preeclampsia (*p* = 0.046) and Mn levels above the 60th percentile (2.11 µg/L) were associated with pre-term delivery (*p* = 0.048) (percentile values are available in Appendix A).

No statistically significant associations were found between the concentrations of the remainder ETE’s and pregnancy complications.

Multivariate logistic regression models were used to assess the risk of pregnancy complications in association with maternal urinary Zn and Mn concentrations, regardless of covariates such as sub-cohort, gestational weight gain, BMI, maternal age, parity, education, and smoking habits (Table 6).

Having urinary Zn levels above the median (50th percentile, P50, 256.9 µg/L) was an independent risk factor for pregnancy complications (adjusted OR [95% CI] of 1.72 [1.03–2.88], *p* = 0.040) and increased by more than 5x the risk of PE (adjusted OR [95% CI] of 5.350 [1.044–27.423], *p* = 0.044). Zn levels below the P5, P10, or P25 (Appendix A) were not associated with PE. Also, no associations were observed between urinary Mn levels below the 35th percentile (P35, 1.02 μg/L) or above the 60th percentile (P60, 2.11 μg/L) and the risk of PE or PTB, respectively (Table 6).

Regarding the neonatal anthropometric profile (Table 7), birth weight and head circumference tended to decrease for higher urinary Mn levels. The association between Mn levels and neonatal head circumference gained statistical significance after adjustment of urinary Mn for creatinine (Appendix A). Additionally, head circumference tended to be higher and birth length tended to be lower for higher Zn levels. However these differences did not reach statistical significance.

Table 8 shows the multivariate logistic regression analysis models created to explore associations between urinary Mn or Zn levels and anthropometry at birth.

The associations previously observed for Mn and anthropometry adequacy were lost after adjustment for sub-cohort, gestational weight gain, education, smoking habits, maternal body mass index (BMI), maternal age, parity, pregnancy complications and newborn’s sex. On the other hand, multivariate logistic regression showed that having Zn levels above the 50th percentile decreased the risk of being born with SGA head circumference (adjusted OR [95% CI] of 0.315 [0.113–0.883], *p* = 0.028), but increased the risk of being born with SGA length (adjusted OR [95% CI] of 2.531 [1.06–6.062], *p* = 0.037).

## 4. Discussion

In the present study, 1st trimester urinary concentrations of Co, Cu, Mn, Mo, and Zn were evaluated in pregnant women from the IoMum cohort, living in Porto and Lisbon regions. Despite the lack of available information on the reference range for urinary concentrations of these ETEs in healthy pregnant populations, the median urinary levels of Co, Cu, Mn, and Zn were within the ranges provided by large international laboratories [40] for the general adult population.

When comparing the present data with other studies in pregnant women, Co, Cu, Mo, and Zn appear to be of the same order of magnitude of those reported among pregnant women during the 1st trimester [41,42,43]. However, the median urinary Mn levels found in the present work were considerably higher than the values reported for other pregnant women during the 1st trimester or for general adult populations [41,44,45,46,47] (details on these ETEs concentrations can be found in Appendix A).

It is important to highlight that the two sub-cohorts of this study had different sociodemographic profiles. This is probably due to the fact that the Porto sub-cohort was recruited in a public hospital and the Lisbon sub-cohort in a private hospital. Accordingly, the educational level was higher in Lisbon sub-cohort, which also had a lower frequency of current smokers and a higher frequency of MV/MMF users, compared with Porto sub-cohort. Furthermore, the Lisbon pregnant women were older and, interestingly, this sub-cohort had a lower frequency of newborns with SGA head circumference. Similar findings have been obtained in other populations [48,49] and may be explained by inherent differences in the socioeconomic status of pregnant women receiving antenatal care at public versus private hospitals.

With regard to the associations between sociodemographic characteristics and ETEs levels, education was positively associated with urinary Mo concentrations. Despite the lack of literature data on Mo levels in pregnancy, a similar positive association between educational level and other ETEs such as iodine has been observed [50,51].

On the contrary, Cu and Mn levels in our study were lower for higher educational levels. A similar association was found by Bocca et al., who reported a negative association between maternal blood (not urine) Mn and educational level (β = −0.56; *p* = 0.001) [52]. These associations can be explained by the use of MV/MMF. In fact, higher educated pregnant women of our cohort were more frequently users of MV/MMF, and MV/MMF use was associated with lower urinary levels of Cu and Mn compared with no use of supplements. Although this may seem paradoxical, MV/MMF used by our participants contained a mixture of microminerals which could compete with each other for intestinal absorption, hampering ETEs’ bioavailability and thus decreasing their urinary excretion. This may be the case of Zn or Fe that were found in the MV/MMF used by the studied population in amounts more than 10 times higher (3.75 to 15 mg of Zn and 5 to 100 mg of Fe per recommended daily dose, Appendix A) compared with Cu and Mn (available at 0.5 to 1 mg or 0.5 to 2 mg, respectively, per recommended daily dose, Appendix A). In fact, Mn, Zn, Cu, and Fe are all transported by (and thus they compete for) the intestinal divalent metal transporter 1 (DMT1) [53,54], which is the main intestinal Mn transporter [55].

On the other hand, increased dietary Zn induces the synthesis of metallothionein by intestinal mucosal cells, an intracellular metal-binding protein with high binding affinity for Cu [16,56], which can result in intracellular retention of Cu and, thus, in a reduction in its absorption [16].

Our data highlight that the use of MV/MMF in pregnancy may not ensure effective provision of certain microminerals that are present in lower amounts in those supplement formulations.

Regarding other lifestyle characteristics, smoking mothers had 2x higher levels of Mn and 1.2x higher levels of Cu and Co, compared with non-smoking mothers. In fact, tobacco contains Cu, Mn, and Co in significant amounts, with a greater abundance of Mn [57,58]. Our findings corroborate other studies in which urinary levels of Co, Cu, and Mn were shown to be higher among smokers compared with non-smokers [59,60,61,62]. Taken together, these findings suggest that urinary Mn, Cu, and Co may be biomarkers of environmental exposure to these ETEs.

In this study, urinary Cu levels were positively associated with BMI. Xu et al. [63] also observed that metabolically unhealthy obese or overweight subjects had higher urinary excretion of Cu (*p* = 0.002) compared with metabolically healthy individuals. Moreover, a recent meta-analysis aiming to assess the relationship between serum Cu status and the anthropometric profile of overweight/obese adults, concluded that serum Cu was significantly higher among obese individuals compared with healthy weight controls (SMD = 0.54 BMI, 95 CI = 0.08 to 1.01) [64]. The mechanisms underlying this association are unclear and should be further explored.

Regarding the studied foods, our results suggest that milk, eggs, fish, and offal can affect ETEs status. Urinary Mn levels among pregnant women who consumed milk at least twice a day were higher (2.39 µg/L) compared with Mn levels among pregnant women consuming milk less frequently. A similar positive association between plasma Mn levels and consumption of dairy products has been observed among the general population by other authors [65]. This can be explained by Mn supplementation for cattle [66,67], a measure implemented in Portugal and in other countries due to its role in reproduction and cattle fetal development [68], which may increase the amount of Mn in cow’s milk. It is important to emphasize that measures of Mn supplementation in cattle must be approached with caution, because urinary Mn levels found in pregnant women consuming at least two 250 mL cups of milk a day (2.39 µg/L) were above the Mn levels observed in pregnant women with pregnancy complications or with low weight newborns.

Regarding other foods, significant inverse associations were found between the consumption of fish and eggs and urinary levels of Cu or Mn. Similarly, Sánchez et al. also observed a negative but weak association between fish and eggs consumption plasma Mn or Cu levels, respectively [65]. In fact, fish and eggs are important dietary sources of Fe [16]. As highlighted above, the absorption of Cu and Mn is affected by the presence of Fe, given the existing competition for DMT1, with Fe prevailing over the other ETEs [54,69].

Consumption of offal was also inversely associated with urinary levels of Mo and Zn. Similar to eggs and fish, offal are a considerable source of Fe [70] and, as noted above and reviewed by Sandstrom [71], Zn absorption is disturbed by the presence of Fe, especially in a solid meal. Interestingly, Cu and Mn levels also tended to be lower for higher frequencies of offal consumption, which reinforces the putative inhibition of intestinal absorption by Fe.

Multivariate logistic models corroborated the association between Zn and pregnancy complications, showing that having urinary Zn levels in the 1st trimester above the median (256.9 µg/L) was an independent risk factor for having at least one pregnancy complication. It is important to emphasize that a five times higher risk of having PE was found for Zn levels above the same cutoff value (256.9 µg/L). In addition, no associations were found between Zn levels below P5, P10, or P25 and PE.

These data may challenge current literature that indicates Zn deficiency as a risk factor for PE [72]. However, despite the large sample size of this study, stratification by pregnancy complications returned a relatively small number of cases of PE (*n* = 16) or PTB (*n* = 25), which must also be taken into consideration when interpreting our results.

Regarding pregnancy complications, lower urinary levels of Mn were found in pregnant women with PE compared with pregnant women without PE (0.99 µg/L and 1.58 µg/L; *p* = 0.046, respectively), and higher levels of Mn were observed in pregnancies complicated with preterm delivery compared with pregnancies with term birth (2.25 µg/L and 1.51 µg/L; *p* = 0.048, respectively). These associations can be explained by at least one of the covariates used in the multivariate logistic regression models: sub-cohort, gestational weight gain, BMI, maternal age, parity, education, or smoking habits.

With regard to neonatal outcomes, an inverse association was observed between maternal urinary Mn levels and adequacy percentiles for two anthropometric parameters: birth weight and birth head circumference. Maternal Mn levels were 26% higher in newborns with small head circumference than in newborns with adequate head circumference (2.12 μg/L and 1.68 μg/L, respectively, *p* = 0.053). Thirty six percent higher Mn levels were also found in low weight compared with normal weight newborns. These observations are also likely to be explained by sub-cohort, gestational weight gain, BMI, maternal age, parity, education, smoking habits, pregnancy complications, or newborn’s sex.

Our data are in line with a nested case-control study reporting that urinary Mn levels above 0.53 μg/L (or at least 1.16 μg/g creatinine) were significantly associated with increased risk for low birth weight [73], even when adjusted for gestational age, family income, maternal BMI, parity, passive smoking and gestational hypertension.

Interestingly, Mn levels observed in our study in pregnant smokers (3.13 µg/L) or in pregnant women who consume two or more cups of milk (≥500 mL) per day (2.39 µg/L) were above the 60th percentile (2.11 µg/L), a cutoff value above which inadequate anthropometry and pregnancy complications were observed. This suggests that smoking or high milk consumption in regions with livestock supplementation with Mn may provide an excessive amount of Mn to pregnant women.

Maternal urinary Zn status also proved to be an important parameter for neonatal anthropometry, since Zn levels above the 50th percentile (256.9 µg/L) were independent protective factors for having small head circumference, while at the same time they increased the risk of having birth length above adequacy.

Birth head circumference is an important parameter for predicting the health status of the newborn and reflects brain development [74]. Despite the lack of literature on the association between maternal Zn levels and this specific anthropometric parameter, our results are in line with the importance of Zn for neurodevelopment [75].

A potential limitation of this study was its observational study design, which makes it difficult to conclude about causality. Another limitation is the fact that our cohort is a convenience sample, not representative of the Portuguese pregnant population, and the analyzed sub-cohorts present significant differences in terms of sociodemographic characteristics and lifestyle. Additionally, urinary levels of Cu, Mn, or Zn may not be good indicators of maternal nutritional status and random spot rather than 24 h urine samples were used in this study. Urinary creatinine is sometimes used to adjust analyte excretion for urine concentration which may not be constant during the day. Despite this, creatinine excretion also depends on age [76], muscle mass [76], nutrition status [4], and BMI [77,78]. In fact, in our sample population creatinine excretion had a weak but statistically significant correlation with maternal age (r = −0.147; *p* < 0.001, Spearmen’s correlation), and it was higher for higher BMI (*p* < 0.001, Kruskal–Wallis) and for weight gain above recommendations for the first trimester (*p* = 0.019, Kruskal–Wallis). In addition, creatinine excretion in our sample varied with food intake such as cheese, offal, and nuts (*p* = 0.039 (Kruskal–Wallis); *p* = 0.033 (Mann–Whitney); *p* = 0.021 (Kruskal–Wallis), respectively). So, results based on urinary ETEs concentration adjusted for creatinine may also be biased by many relevant variables. However, in our study, adjustment of urinary ETEs concentration for creatinine excretion (Appendix A) did not substantially change the main conclusions drawn.

This study also has important strengths. First, it encompasses a large sample size, which dampens the impact of inter-individual variations, being fundamental for greater accuracy in the statistical analysis. Second, as far as we know, this is the first study in Portugal that characterized a relevant number of ETEs in pregnant women. Third, this study used a gold standard method for the quantification of ETEs in urine, which allows comparisons with other ICP-MS-based studies.

## 5. Conclusions

In this work we characterized the urinary levels of five ETEs in Portuguese pregnant women. Our results show that smoking can be a source of Mn exposure at levels that are associated with inadequate anthropometry and pregnancy complications. On the other hand, taking MV/MM food supplements may not ensure the effective supply of Cu or Mn, which are present in much lower amounts in commercial formulations compared with Zn or Fe, which compete with Cu and Mn for intestinal absorption.

Regarding gestational and neonatal outcomes, high Mn levels associated with PTB and low Mn levels associated with PE, but these associations were dependent on several covariates.

Finally, having urinary Zn levels above the 50th percentile in the first trimester of pregnancy was an independent risk factor for PE, although it was a protective factor for being born with small head circumference.

Taken together, the results obtained in this study may provide valuable information for public health policies related to prenatal nutrition, while informing future efforts to define urinary reference intervals for ETEs in pregnant women.

## Figures and Tables

**Figure 1 biology-12-01351-f001:**
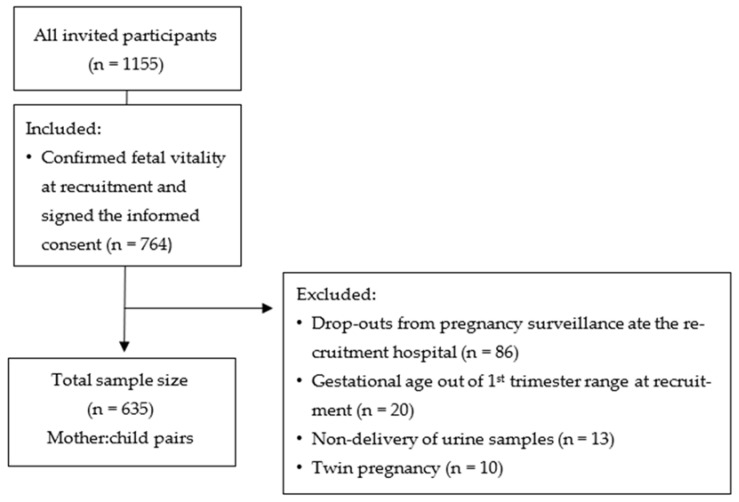
Flowchart of the participant recruitment process.

**Figure 2 biology-12-01351-f002:**
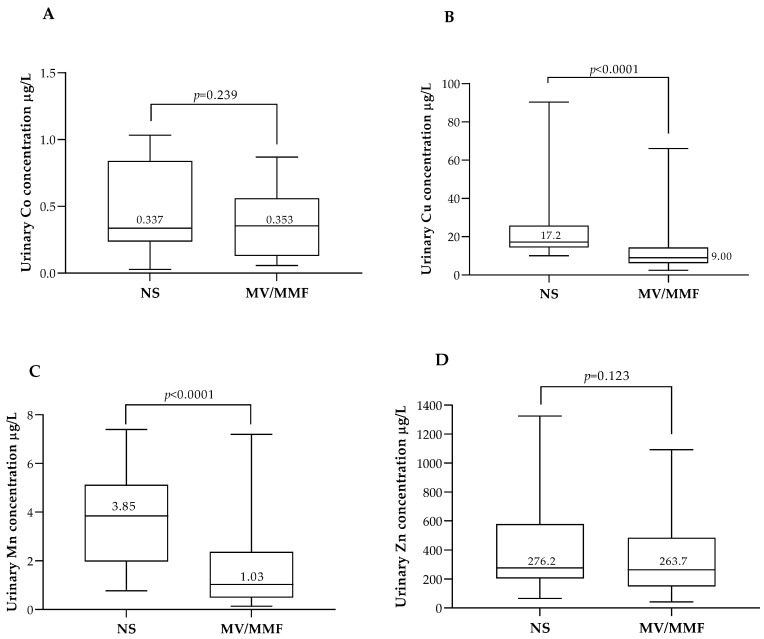
Maternal urinary concentrations of Co (**A**), Cu (**B**), Mn (**C**), and Zn (**D**) in the 1st trimester according to food supplement use. In each box, the median value (in µg/L) is indicated above the corresponding central horizontal line. *p* values (Mann–Whitney) are indicated in each panel. MV/MMF: multivitamin/multimineral formulation users; NS: non-users of supplements.

**Figure 3 biology-12-01351-f003:**
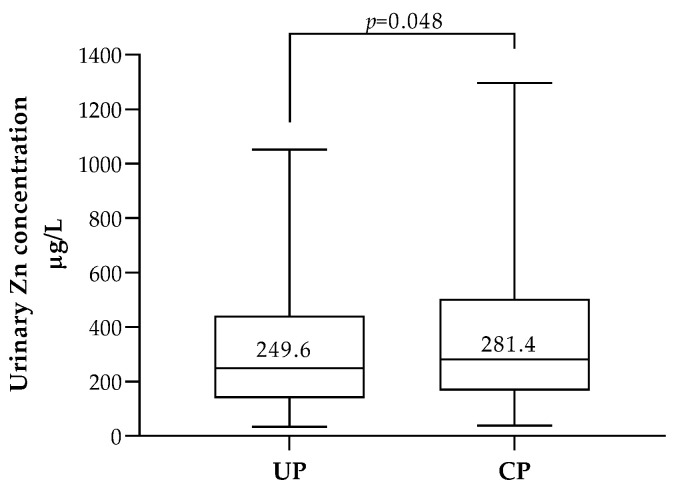
First-trimester maternal urinary Zn concentrations in uncomplicated (UP, *n* = 476) and complicated (CP, *n* = 120) pregnancies. In each box, the median value (in µg/L) is indicated above the corresponding central horizontal line. *p* value (Mann–Whitney) are indicated.

**Figure 4 biology-12-01351-f004:**
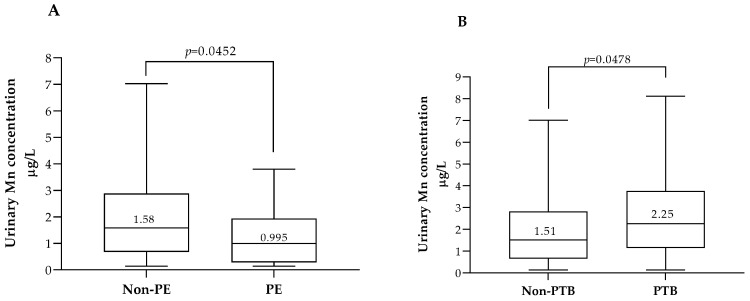
First-trimester maternal urinary Mn concentrations according to the type of pregnancy complication. (**A**), Mn concentrations in pregnancies with preeclampsia (PE, *n* = 16) or pregnancies without preeclampsia (Non-PE, *n* = 579). (**B**), Mn concentrations in pregnancies with preterm delivery (PTB, *n* = 25) or non-preterm delivery (Non-PTB, *n* = 571). In each box, the median value (in µg/L) is indicated above the corresponding central horizontal line. *p* values (Mann–Whitney) are indicated in each panel.

**Table 1 biology-12-01351-t001:** Biological functions and dietary characteristics of ETEs.

ETE	Enzymes Using ETE as Cofactors	Biological Functions	Food Sources	RDA for Pregnant Women	Refs
Co	Methionine synthase L-methyl-malonylCoA mutase	Component of Vitamin B12 (Vit B12, cobalamin); metabolism of folates and purines;synthesis of methionine	Fresh cereals and green vegetables (0.2–0.6 µg Co/g dry mass)	Not available for Co; Vit B12: 2.6 µg/day, which corresponds to about 0.1 µg of Co	[6,7,8,9]
Cu	Oxireductases (e.g., copper/zinc superoxide dismutase and cytochrome oxidase)	Cellular respiration; antioxidant defense; neuropeptide synthesis	Shellfish, nuts, sunflower seeds, cocoa, bran cereals, whole grain products, beef liver and other organ meats	1000 µg/day	[10,11,12,13,14,15,16]
Mn	Metalloenzymes (e.g., arginase, glutaminase synthetase and phosphoenolpyruvate decarboxylase) Oxireductases (manganese superoxide dismutase)	Carbohydrate, protein, amino acid and lipid metabolism; antioxidant defense	Whole grains, legumes, nuts, seeds, tea, wheat, brown rice, and spinach	Not available	[16,17,18,19]
Mo	Sulfite oxidase, xanthine oxidoreductase, aldehyde oxidase, and mitochondrial amidoxime-reducing component	Metabolism of sulfur amino acids and heterocyclic compounds (e.g., purines and pyridines) Production of uric acid and in detoxification	Beans, oat, rice, nuts, and dark-leafy vegetables	50 μg/day	[16,20,21]
Zn	Over 300 metalloenzymes (e.g., carbonic anhydrase and alkaline phosphatases and) Oxireductases (e.g., copper/zinc superoxide dismutase)	Protein synthesis; antioxidant defense	Meat, seafood, dairy products, eggs, seeds, and nuts	11 mg/day	[16,22,23,24]

Legend: Co: cobalt; Cu: copper; ETEs: essential trace elements; Mn: manganese; Mo: molybdenum; Zn: zinc.

**Table 2 biology-12-01351-t002:** Sociodemographic and clinical characteristics of the study sample.

Characteristics	Total	Porto	Lisbon	*p*
	(*n* = 635)	(*n* = 337)	(*n* = 298)	
Age, *n*	631	336	295	
Mean ± SD, years	33 ± 5	32 ± 5	34 ± 5	<0.001 ^a^
Gestational age at recruitment, *n*	635	337	298	
Median (P25; P75), weeks	12 (12; 13)	12 (12; 13)	12 (12; 12)	0.265 ^b^
Gestational age at delivery, *n*	602	336	298	
Median (P25; P75), weeks	39 (38; 40)	39 (38; 40)	39 (38; 40)	0.763 ^b^
Sub-cohort, *n* (%)				
Porto	337 (53)	nd	nd	nd
Lisbon	298 (47)			
Education level, *n* (%)				
Low (≤ 9 years)	155 (25)	154 (48)	1 (0)	<0.001 ^c^
Medium (10 to 12 years)	181 (29)	122 (38)	59 (20)	
University (≥ 13 years)	282 (46)	47 (15)	235 (80)	
Smoking habits, *n* (%)				
Non-smoker	443 (70)	232 (70)	211 (71)	0.001 ^c^
Former smoker	132 (21)	60 (18)	72 (24)	
Smoker	56 (9)	42 (13)	14 (5)	
Use of nutritional supplements, *n* (%)				
NS	23 (4)	20 (7)	3 (1)	<0.001 ^c^
MV/MMF	249 (46)	88 (33)	161 (59)	
SIF	269 (50)	160 (60)	109 (40)	
Pre-pregnancy BMI, *n* (%)				
Underweight	35 (6)	20 (6)	15 (5)	0.846 ^c^
Normal weight	400 (63)	209 (62)	191 (64)	
Overweight	114 (18)	59 (18)	55 (19)	
Obese	85 (14)	48 (14)	37 (12)	
Total weight variation, *n* (%) *				
below adequacy	96 (20)	58 (21)	38 (19)	0.841 ^c^
adequate	163 (34)	93 (33)	70 (36)	
above adequacy	217 (46)	129 (46)	88 (45)	
1st trimester weight variation, *n* (%) ^#^				
Below adequacy	132 (21)	72 (22)	60 (21)	0.809 ^c^
Adequate	189 (31)	97 (30)	92 (32)	
Above adequacy	293 (48)	158 (48)	135 (47)	
Pregnancy complications, *n* (%)				
No	476 (80)	248 (83)	228 (77)	0.041 ^c^
Yes	120 (20)	50 (17)	70 (24)	
Preeclampsia	16 (3)	6 (2)	10 (3)	0.314 ^c^
Gestational diabetes	56 (9)	23 (8)	33 (11)	0.164 ^c^
Fetal growth restriction	13 (2)	11 (4)	2 (1)	0.011 ^c^
Infection	3 (1)	1 (0)	2 (1)	1.000 ^d^
Preterm birth	25 (4)	18 (6)	7 (2)	0.025 ^c^
Parity, *n* (%)				
Nulliparous	314 (51)	168 (52)	146 (49)	0.671 ^c^
Primiparous	256 (41)	132 (41)	124 (42)	
Multiparous	49 (8)	23 (7)	26 (9)	
Sex, *n* (%)				
Male	313 (49)	164 (49)	149 (50)	0.737 ^c^
Female	322 (51)	173 (51)	149 (50)	
Birth weight, *n*	635	337	298	
Mean ± SD, grams	3207 ± 433	3158 ± 460	3263 ± 394	0.002 ^a^
SGA, *n* (%)	34 (5)	24 (7)	10 (3)	0.088 ^c^
AGA, *n* (%)	580 (92)	303 (90)	277 (93)	
LGA, *n* (%)	20 (3)	9 (3)	11 (4)	
Birth head circumference, *n*	612	326	286	
Median (P25; P75), cm	34.0 (33.5; 35.0)	34.0 (33.0; 35.0)	34.5 (33.7; 35.5)	<0.001 ^b^
SGA, *n* (%)	32 (5)	25 (8)	7 (2)	0.002 ^c^
AGA, *n* (%)	523 (86)	278 (86)	245 (86)	
LGA, *n* (%)	56 (9)	22 (7)	34 (12)	
Birth length, *n*	624	337	287	
Mean ± SD, cm	49.0 ± 2.0	49.1 ± 2.2	48.8 ± 1.7	0.029 ^a^
SGA, *n* (%)	49 (8)	20 (6)	29 (10)	0.031 ^c^
AGA, *n* (%)	570 (92)	312 (93)	258 (90)	
LGA, *n* (%)	4 (1)	4 (1)	0 (0)	

Legend: AGA: adequate for gestational age; LGA: large for gestational age; MV/MMF: multivitamin/multimineral formulations; nd: not determined; NS: non-users of supplements; SGA: small for gestational age; SIF: single ingredient formulations. ^a^ Student’s *t*-test. ^b^ Mann–Whitney. ^c^ Pearson chi-square. ^d^ Fisher’s exact test. * Total weight variation corresponds to the difference between weight at term and weight 6 months before pregnancy. # Weight variation during the 1st trimester corresponds to the difference between weight at recruitment day and weight 6 months before pregnancy. Missing data interval, *n* (%): 0–159 (0–29). The maximum number of missing data refers to lack of information on total weight variation.

**Table 3 biology-12-01351-t003:** Urinary ETEs concentrations (*n* = 635) in pregnant women.

ETEs (μg/L)	<LOD *		Min	P5	P25	Median	P75	P95	Max
	*n*	%							
Co	113	18	0.03	0.03	0.13	0.31	0.53	1.08	3.92
Cu	0	0	1.01	3.70	7.00	11.1	17.8	59.6	328.9
Mn	48	8	0.14	0.14	0.71	1.67	3.05	5.95	18.6
Mo	0	0	3.22	8.41	22.1	39.3	63.6	107.7	468.7
Zn	1	0	8.57	53.0	145.8	256.9	446.8	839.4	3620.9

Legend: Co: cobalt; Cu: copper; ETEs: essential trace elements; LOD: limit of detection; Mn: manganese; Mo: molybdenum; P: percentile; Zn: zinc. * Absolute number and percentage of results below the LOD.

**Table 4 biology-12-01351-t004:** Sociodemographic characteristics of pregnant women and urinary levels of ETEs.

Characteristics	*n*	%	Co (μg/L)			Cu (μg/L)			Mn (μg/L)			Mo (μg/L)			Zn (μg/L)		
			Median	(P25; P75)	*p*	Median	(P25; P75)	*p*	Median	(P25; P75)	*p*	Median	(P25; P75)	*p*	Median	(P25; P75)	*p*
Sub-cohort																	
Porto	337	53	0.27	(0.08; 0.51)	<0.001 ^a^	15.5	(10.0; 26.3)	<0.001 ^a^	2.83	(1.89; 4.30)	<0.001 ^a^	36.8	(21.1; 58.5)	0.004 ^a^	258.9	(148.8; 446.0)	0.799 ^a^
Lisbon	298	47	0.35	(0.17; 0.55)		8.36	(5.81; 11.5)		0.73	(0.33; 1.27)		45.3	(23.1; 68.0)		253.5	(141.2; 452.8)	
Education level																	
Low (≤9 years)	155	25	0.26	(0.12; 0.48)	0.158 ^b^	17.0	(9.90; 28.1)	<0.001 ^b^	2.99	(1.9; 4.31)	<0.001 ^b^	36.3	(20.5; 51.0)	0.007 ^b^	276.2	(149.3; 487.9)	0.207 ^b^
Medium (10 to 12 years)	181	29	0.32	(0.10; 0.56)		12.0	(7.89; 18.8)		2.03	(1.00; 3.62)		37.8	(21.1; 59.8)		275.9	(163.2; 439.6)	
University (≥13 years)	282	46	0.33	(0.15; 0.54)		9.02	(6.18; 12.9)		0.90	(0.44; 1.72)		45.6	(23.2; 70.4)		245.0	(132.3; 441.6)	
Smoking habits																	
Non-smoker	443	70	0.29	(0.10; 0.52)	0.076 ^b^	11.0	(6.84; 18.0)	0.047 ^b^	1.61	(0.68; 2.92)	<0.001 ^b^	40.1	(21.2; 64.6)	0.582 ^b^	256.9	(139.8; 437.1)	0.502 ^b^
Former smoker	132	21	0.34	(0.17; 0.55)		10.0	(6.86; 16.7)		1.45	(0.68; 2.60)		39.5	(26.2; 63.0)		257.9	(162.1; 481.1)	
Smoker	56	9	0.35	(0.19; 0.53)		12.8	(9.84; 24.4)		3.13	(1.11; 4.38)		36.7	(22.8; 50.9)		278.9	(145.2; 532.5)	
Pre-pregnancy BMI																	
Underweight	35	6	0.23	(0.15; 0.40	0.336 ^b^	9.26	(6.73; 14.5)	0.027 ^b^	2.33	(0.59; 3.09)	0.349 ^b^	33.2	(20.8; 59.8)	0.236 ^b^	231.8	(161.0; 385.8)	0.062 ^b^
Normal weight	400	63	0.32	(0.13; 0.54)		10.7	(6.77; 17.3)		1.66	(0.69; 3.01)		40.1	(22.4; 64.7)		252.2	(139.3; 436.4)	
Overweight	114	18	0.27	(0.09; 0.50)		11.6	(7.44; 17.0)		1.44	(0.63; 2.63)		35.5	(19.8; 53.5)		256.6	(138.6; 495.7)	
Obese	85	13	0.33	(0.18; 0.57)		14.3	(8.37; 25.0)		1.72	(1.01; 3.58)		41.1	(23.7; 65.3)		313.8	(188.4; 531.7)	
1st trimester weight variation																	
Below adequacy	132	21	0.27	(0.10; 0.45)	0.020 ^b^	13.2	(7.08; 23.5)	0.008 ^b^	1.70	(0.75; 2.82)	0.485 ^b^	32.5	(20.2; 64.6)	0.057 ^b^	250.0	(149.2; 455.8)	0.219 ^b^
Adequate	189	31	0.28	(0.09; 0.49)		9.82	(6.37; 15.7)		1.52	(0.64; 2.99)		35.2	(21.0; 58.6)		241.0	(139.4; 410.1)	
Above adequacy	293	48	0.33	(0.17; 0.57)		11.2	(7.65; 17.3)		1.76	(0.75; 3.42)		44.0	(24.8; 64.5)		267.7	(148.2; 472.0)	

Legend: Co: cobalt; Cu: copper; ETEs: essential trace elements; Mn: manganese; Mo: molybdenum; Zn: zinc. ^a^ Mann–Whitney. ^b^ Kruskal–Wallis. Missing data interval, *n* (%): 0–159 (0–29).

**Table 5 biology-12-01351-t005:** Urinary concentrations of ETEs in the 1st trimester by frequency of intake of milk, eggs, fish, seafood, and offal.

Food Intake	*n*	%	Co (μg/L)			Cu (μg/L)			Mn (μg/L)			Mo (μg/L)			Zn (μg/L)		
			Median	(P25; P75)	*p*	Median	(P25; P75)	*p*	Median	(P25; P75)	*p*	Median	(P25; P75)	*p*	Median	(P25; P75)	*p*
Milk																	
<3 times a month	178	28.4	0.31	(0.10; 0.52)	0.677 ^a^	10.1	(7.14; 17.5)	0.686 ^a^	1.26	(0.53; 2.58)	0.002 ^a^	39.6	(23.3; 60.9)	0.584 ^a^	266.19	(153.9; 463.7)	0.570 ^a^
1 a 6 times a week	157	25.0	0.32	(0.15; 0.52)		11.6	(6.84; 17.1)		1.56	(0.77; 2.72)		44.8	(20.8; 65.5)		263.87	(145.0; 504.2)	
1 time a day	216	34.4	0.33	(0.14; 0.56)		11.6	(7.65; 18.1)		1.77	(0.75; 3.29)		38.7	(22.6; 64.37)		253.46	(160.4; 407.4)	
≥2 times a day	76	12.1	0.28	(0.12; 0.49)		10.3	(6.01; 19.8)		2.39	(0.97; 4.00)		34.3	(21.4; 58.4)		225.38	(119.3; 466.3)	
Eggs																	
<3 times a month	126	20.2	0.32	(0.15; 0.53)	0.993 ^a^	11.7	(7.87; 20.5)	0.015 ^a^	2.11	(1.03; 3.68)	<0.001 ^a^	39.6	(22.5; 59.8)	0.721 ^a^	257.01	(141.8; 470.2)	0.665 ^a^
1 a 3 times a week	410	65.6	0.31	(0.13; 0.53)		11.5	(7.15; 17.5)		1.67	(0.73; 2.99)		40.2	(22.3; 65.3)		263.81	(153.2; 443.9)	
≥4 times a week	89	14.2	0.32	(0.12; 0.56)		9.47	(6.15; 15.3)		1.02	(0.47; 2.06)		38.7	(22.6; 59.8)		242.44	(137.6; 472.1)	
Fish																	
<3 times a month	71	11.6	0.32	(0.14; 0.55)	0.709 ^a^	12.6	(8.46; 19.8)	0.041 ^a^	1.83	(0.94; 3.47)	0.005 ^a^	41.1	(20.1; 61.5)	0.722 ^a^	281.99	(181.0; 479.7)	0.313 ^a^
1 a 3 times a week	407	66.6	0.31	(0.12; 0.52)		11.0	(7.00; 17.6)		1.67	(0.75; 2.85)		39.0	(22.2; 63.6)		249.44	(139.4; 436.7)	
≥4 times a week	133	21.8	0.33	(0.16; 0.56)		9.61	(6.57; 15.5)		1.04	(0.48; 2.68)		40.2	(22.7; 65.1)		266.29	(156.3; 471.6)	
Seafood																	
do not eat Seafood at all	180	60.8	0.35	(0.17; 0.56)	0.597 ^b^	8.32	(5.87; 11.7)	0.630 ^b^	0.75	(0.38; 1.28)	0.181 ^b^	49.2	(24.7; 74.3)	0.029 ^b^	264.49	(154.9; 467.6)	0.086 ^b^
≥1 time a month	116	39.2	0.33	(0.16; 0.52)		8.44	(5.72; 11.1)		0.66	(0.31; 1.18)		40.0	(20.3; 61.3)		242.13	(132.0; 411.6)	
Offal																	
do not eat Offal at all	228	80.0	0.35	(0.17; 0.56)	0.092 ^b^	8.46	(5.98; 11.7)	0.067 ^b^	0.76	(0.36; 1.31)	0.153 ^b^	47.2	(24.3; 67.7)	0.046 ^b^	267.97	(148.7; 489.9)	0.047 ^b^
≥1 time a month	57	20.0	0.27	(0.12; 0.48)		7.52	(5.35; 9.92)		0.60	(0.31; 1.05)		33.8	(15.3; 64.2)		219.76	(123.9; 355.4)	

Legend: Co: cobalt; Cu: copper; ETEs: essential trace elements; Mn: manganese; Mo: molybdenum; Zn: zinc. The respective portions of the food groups are indicated in Appendix A; for Seafood and Offal the data refer exclusively to the Lisbon sub-cohort. ^a^ Kruskal–Wallis. ^b^ Mann–Whitney. Missing data interval, *n* (%): 8–350 (1–55). The maximum number of missing data refers to lack of information on Seafood and Offal intake as explained in Section 2.2.

**Table 6 biology-12-01351-t006:** Multivariate logistic regression analysis of the risk of pregnancy complications, preeclampsia and preterm delivery in association with Zn or Mn levels.

	*n*	Pregnancy Complications ^a^			PE			PTB		
		aOR ^b^	(95% CI)	*p*	aOR ^b^	(95% CI)	*p*	aOR ^b^	(95% CI)	*p*
Zinc, μg/L										
≤P50 (256.9 μg/L)	227	1			1			1		
>P50 (256.9 μg/L)	229	1.72	(1.03–2.88)	0.040	5.35	(1.04–27.42)	0.044	2.06	(0.615–6.89)	0.241
Mn, μg/L										
≤P35 (1.02 μg/L)	154	nd			1.86	(0.397–8.72)	0.430	nd		
>P35 (1.02 μg/L)	302	nd			1			nd		
≤P60 (2.11 μg/L)	270	nd			nd			1		
>P60 (2.11 μg/L)	186	nd			nd			0.958	(0.263–3.48)	0.948

Legend: aOR: adjusted odds ratio; Mn: manganese; nd: not determined; PE: preeclampsia; PTB: preterm delivery; Zn: zinc. ^a^ Preeclampsia, gestational diabetes, fetal growth restriction, infection, preterm birth, spontaneous fetal loss, and others. ^b^ Adjusted for sub-cohort, gestational weight gain, education, smoking habits, maternal body mass index (BMI), maternal age and parity.

**Table 7 biology-12-01351-t007:** Association between first trimester urinary ETEs levels and neonatal outcomes.

Neonatal Outcomes	*n*	%	Co (μg/L)			Cu (μg/L)			Mn (μg/L)			Mo (μg/L)			Zn (μg/L)		
			Median	(P25; P75)	*p*	Median	(P25; P75)	*p*	Median	(P25; P75)	*p*	Median	(P25; P75)	*p*	Median	(P25; P75)	*p*
Birth weight adequacy																	
SGA	34	5	0.30	(0.12; 0.45)	0.150 ^a^	10.2	(6.91; 14.7)	0.602 ^a^	2.26	(1.12; 3.30)	0.071 ^a^	35.6	(22.8; 55.7)	0.816 ^a^	209.8	(111.9; 352.8)	0.189 ^a^
AGA	580	91	0.32	(0.14; 0.55)		11.2	(7.00; 18.3)		1.66	(0.69; 3.06)		39.8	(21.9; 64.2)		259.0	(148.4; 458.7)	
LGA	20	3	0.24	(0.06; 0.45)		10.1	(6.42; 15.7)		1.06	(0.51; 2.26)		37.4	(22.8; 53.8)		239.8	(149.5; 360.3)	
Birth head circumference adequacy																	
SGA	32	5	0.25	(0.08; 0.48)	0.371 ^a^	9.4	(6.74; 14.7)	0.312 ^a^	2.12	(1.35; 3.05)	0.053 ^a^	37.0	(19.3; 60.7)	0.239 ^a^	195.2	(115.7; 314.5)	0.088 ^a^
AGA	523	86	0.32	(0.14; 0.54)		11.4	(7.02; 18.6)		1.68	(0.72; 3.08)		38.7	(22.3; 62.0)		259.5	(148.5; 458.2)	
LGA	56	9	0.35	(0.14; 0.52)		10.9	(6.72; 17.0)		1.09	(0.48; 2.67)		46.4	(28.0; 72.2)		281.7	(170.8; 470.9)	
Birth length adequacy																	
SGA	49	8	0.36	(0.16; 0.51)	0.582 ^b^	11.8	(8.73; 17.5)	0.945 ^b^	1.71	(0.64; 3.36)	0.885 ^b^	45.8	(25.7; 65.3)	0.210 ^b^	316.8	(216.6; 481.2)	0.152 ^b^
AGA	570	91	0.31	(0.13; 0.54)		11.2	(6.91; 18.1)		1.67	(0.74; 3.03)		38.7	(21.7; 62.5)		252.3	(146.0; 443.8)	
LGA	4	1	nd	nd		nd	nd		nd	nd		nd	nd		nd	nd	

Legend: AGA: adequate for gestational age; ETEs: essential trace elements; LGA: large for gestational age; nd: not determined due to small sample size; SGA: small for gestational age. ^a^ Kruskal–Wallis. ^b^ Mann–Whitney. Missing data interval, *n* (%): 1–24 (0.2–4).

**Table 8 biology-12-01351-t008:** Multivariate logistic regression analysis of the risk of anthropometric inadequacy of the newborn in association with urinary levels of Zn or Mn.

	BW SGA				BW LGA				BHC SGA				BHC LGA				BL SGA			
	*n*	aOR ^a^	(95% CI)	*p*	*n*	aOR ^a^	(95% CI)	*p*	*n*	aOR ^a^	(95% CI)	*p*	*n*	aOR ^a^	(95% CI)	*p*	*n*	aOR ^a^	(95% CI)	*p*
Zinc, μg/L																				
≤P50 (256.86 μg/L)	nd				nd				202	1			205	1			219	1		
>P50 (256.86 μg/L)	nd				nd				202	0.315	0.113–0.883	0.028	219	1.06	0.528–2.12	0.871	225	2.53	1.06–6.06	0.037
Mn, μg/L																				
≤P35 (1.02 μg/L)	nd				150	0.331	0.061–1.793	0.200	nd				143	0.992	0.431–2.28	0.985	nd			
>P35 (1.02 μg/L)	nd				283	1			nd				281	1			nd			
≤P60 (2.11 μg/L)	261	1			nd				231	1			nd				nd			
>P60 (2.11 μg/L)	190	0.551	0.181–1.68	0.295	nd				173	1.27	0.438–3.70	0.658	nd				nd			

Legend: aOR: adjusted odds ratio; AGA: adequate for gestational age; BW: birth weight; BHC: birth head circumference; BL: birth length; LGA: large for gestational age; Mn: manganese; nd: not determined; SGA: small for gestational age; Zn: zinc. ^a^ Adjusted for sub-cohort, gestational weight gain, education, smoking habits, maternal body mass index (BMI), maternal age, parity, pregnancy complications, and newborn sex.

## Data Availability

The data sets generated during and/or analyzed during the current study are available from the corresponding author on reasonable request.

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
