# Peer review of "Essential Trace Elements Status in Portuguese Pregnant Women and Their Association with Maternal and Neonatal Outcomes: A Prospective Study from the IoMum Cohort"

_biology, 2023, doi:10.3390/biology12101351_

Round 1
Reviewer 1 Report
BIOLOGY: Essential trace elements status in Portuguese pregnant women 2 and their association with maternal and neonatal outcomes: a 3 prospective study from the IoMum cohort.
This is an interesting paper with some important findings that should be published. However, it provides far more detail than is needed to reach the conclusions that are reached and the extra detail detracts from appreciating the important results. In addition, there are some language imperfections that can easily be corrected.
line 55 and 136 What is a "random urine sample"? does this mean the time of day varied, time in the pregnancy varied?
Do relationships take into account other ETEs or socio-demographic factors?
Figure 1. This is not clear. I subtracted 41 + 21+ 17 from 765 and obtained a sample size of 686, not 635. Please provide a very clear diagram showing the reduction of sample size, step by step.
Lines 216-217. 69% had inadequate weight gain in the first trimester and 48% had more gain than recommended. 69 + 48 = more than 100%. Something is not clear in this sentence.
Table 2: What is "Adequacy of total weight variation, n (%)* " Is this meant to be Adequate weight gain during pregnancy?
Table 2: There is repetition in the names of the variables.
Neonates sex can be changed to Sex because sex can only refer to the neonates as all the mothers are female.
Neonatal birth weight is always at birth. Therefore, Neonatal is superfluous here Neonatal Head Circumference. Neonatal is superfluous here
The abbreviations for nutritional supplementation require more explanation and should be a footnote in the table with a mark, such as * in the table to lead the reader to the footnote.
Somewhere early in the paper, the abbreviation ETE should be spelled out.
You used a formula to replace values below the limit of detection. In table 3 is the minimum value one that is based on those substitutions or is it the minimum value above the LOD? Please clarify in the paper or the table.
Lines 339-342: Verbs are missing from these sentences: " Additionally, lower or higher level of urinary Mn associated with specific complications (Figure 4). Urinary Mn levels below the 35th percentile (1.02 μg/L) associated with preeclampsia (p = 0.046) and above the 60th percentile (2.11 μg/L) 341 associated with pre-term delivery (p = 0.048).
Table 4 provides more information than is needed because the significant associations are also described in detail. Table 4 may be eliminated if some additional text states which ETEs did not differ by socioeconomic variables.
Table 5 presents 45 comparisons but only 9 a noteworthy (show significant differences). This table can be eliminated if the authors add to the text statements of which ETEs are related to dietary intake measures.
Table 6. Make the explanation for the abbreviation "nd" more obvious. Each abbreviation in the table should have its own footnote. Explain why the relationship was Not Determined.
Line 369: this statement is too strong, " consistent trend of inverse association was observed between urinary Mn levels and adequacy of birth weight for gestational age," because the differences are not statistically significant. Perhaps referring to this as a "suggestive trend".
The language could be clearer if the term "adequacy" was dropped. The relationships depicted are with size for gestational age. This is a simpler term and does not imply any qualitative assessment which the term "adequacy" does.
Line 384-5, 386: Again, the term "adequacy" is misleading and unnecessary. This sentence, " Table 8 shows the multivariate logistic regression analysis models created to explore associations between urinary Mn or Zn levels and adequacy of anthropometry at birth." could be rewritten as "Table 8 shows the multivariate logistic regression analysis models created to explore associations between urinary Mn or Zn levels and anthropometry at birth." The meanings of both sentences are the same but the second is shorter and clearer.
Line 386: If the associations reported are no longer significant when including relevant sociodemographic variables in a multivariate model, then those associations that are no longer significant could be deleted from the text or given far less space in the paper.
Line 415: word missing, " when compared the adult"
Line 415-417: I do not recall seeing a comparison of the data from the study population with the values for the adult population. Therefore, this statement, "with median ETEs concentrations higher than those found in adult general population for Co [59], Cu [59-61], Mn [60-62], Mo [59-61], and Zn [60]." is an opinion without substantiation. A table comparing the values would be needed to substantiate this assertion.
Lines 462-464: Where in the paper are the results that show the following relationship? "Mn levels found in pregnant smokers in our study (3.13 μg/L) were above the 60th percentile, a cutoff that was associated with inadequate anthropometry and pregnancy 463 complications, respectively."
English language usage is good but some imperfections are distracting or contribute to ambiguity and should be corrected.
Author Response
We greatly appreciate the insightful comments from the Reviewer on the manuscript submitted. To access the answers requested, please consult the document provided.
Thank you.
Best regards,

Reviewer 2 Report
The manuscript „Essential trace elements status in Portuguese pregnant women and their association with maternal and neonatal outcomes: a prospective study from the IoMum cohort.” Shows very interesting data in an impressive cohort. A cohort of 635 mother:child pairs very selected and observed regarding selected essential trace element concentrations in urine and if those impact the birth outcome. The introduction is clear and especially the table is great as it contains a lot of info about the trace elements that are presented short and precise. In the material and method section the study is well explained, and the flowchart is helpful to understand the study. In line 136, the authors state that “random urine sample was collected”. And here lies the biggest issue in the whole study. Firstly, urine is no good biomarker for trace element status, if it is used then it would be more reliable to use 24-hour urine. Secondly, there should be any normalization parameters, like creatinine concentration. Please discuss on this issue in the manuscript. The analytical method should be discussed more detailed, which ICP-MS, sample preparation, where LODs and LOQs measured daily or only once etc. There is no description in the material and method section about the intake data, those data suddenly appear in table 5 without any additional information. Section 2.5 is clear, as here a big cohort was observed, the use of Shapiro wilk test could be more robust. In line 261, the authors write about positive association, but here only significances were compared. So please consider the use of the terms association and significances, as an association needs an correlation analysis. In figure 2 IF is introduced, but at this point it is not clear which single ingredient was in the formula. With the presentation of data in that way it implies that, the respective trace element has been applied. Please explain here more precisely. Why are only the elements Co, Cu, Mn, and Zn shown in figure 2, where is Mo? Which significance test was used here, and what do the stars represent. For figure 3 it is the same, where are the other trace elements? And define the stars here as well. Table 7 is highly interesting, but it is still no association it is a comparison of the significances again. In table 8 some values are displayed in bold. Why? Please make this clear. The discussion too long and should be more precise and shortened. Citation 41 is not complete. To conclude, in my opinion this manuscript should be rated with minor revision.
Author Response
We greatly appreciate the insightful comments from the Reviewer on the manuscript submitted. To access the answers requested, please consult the document submitted.
Thank you.
Best regards,

Round 2
Reviewer 1 Report
You have been very responsive to my comments on the original submission.
Reviewer 2 Report
All comments have benn aswered. Thank you!
Maybe the Journal should check for English proof.